# Biomolecular Corona Stability in Association with Plasma Cholesterol Level

**DOI:** 10.3390/nano12152661

**Published:** 2022-08-03

**Authors:** Duong N. Trinh, Meda Radlinskaite, Jack Cheeseman, Gunter Kuhnle, Helen M. I. Osborn, Paula Meleady, Daniel I. R. Spencer, Marco P. Monopoli

**Affiliations:** 1Department of Chemistry, Royal College of Surgeons in Ireland, RCSI University of Medicine and Health Sciences, D02 YN77 Dublin, Ireland; ngocduongtrinh@rcsi.ie (D.N.T.); c18307936@mytudublin.ie (M.R.); 2School of Physics & Clinical & Optometric Sciences, Technological University Dublin, D07 EWV4 Dublin, Ireland; 3Ludger Ltd., Culham Science Centre, Abingdon OX14 3EB, UK; jack.cheeseman@ludger.com (J.C.); daniel.spencer@ludger.com (D.I.R.S.); 4School of Pharmacy, University of Reading, Whiteknights, Reading RG6 6AD, UK; h.m.i.osborn@reading.ac.uk; 5Department of Food and Nutritional Sciences, University of Reading, Whiteknights, Reading RG6 6AH, UK; g.g.kuhnle@reading.ac.uk; 6School of Biotechnology, Dublin City University, D09 W6Y4 Dublin, Ireland; paula.meleady@dcu.ie

**Keywords:** protein corona, lipoproteins, colloidal stability, silica nanoparticles

## Abstract

Biomolecular corona is spontaneously formed on the surface of nanoparticles (NPs) when they are in contact with biological fluids. It plays an important role in the colloidal stability of NPs, which is of importance for most of their medical applications and toxicity assessment. While typical studies use either blood plasma or serum from a pooled biobank, it is unclear whether differences in the media, such as cholesterol level or protein concentration, might affect the NP colloidal stability and corona composition. In this study, the silica corona was prepared at particularly low plasma concentrations (3%, *v*/*v*–1.98 mg/mL) to identify the critical roles of the protein mass/NP surface ratio and the level of plasma cholesterol on the corona protein pattern and particle stability. While depending on the plasma dilution factor, the corona protein composition could be controlled by keeping the protein/NP constant. The NP colloidal stability was found to strongly correlate with the level of cholesterol in human plasma, particularly due to the high enrichment of high-density lipoprotein (HDL) and low-density lipoprotein (LDL) in the corona. A cohort study on plasma samples from individuals with known cholesterol levels was performed to highlight that association, which could be relevant for all corona systems enriched with the LDL.

## 1. Introduction

With high surface energy, nanoparticles (NPs) incubated in biological fluids are quickly covered by surrounding biomolecules, mainly proteins, to form a new entity called the ‘biomolecular corona’ [1,2,3]. As the corona bestows NPs with a new biological identity, controlling the formation of the corona could provide a new strategy for nano-therapeutics, as well as for biomarker discovery [4,5,6,7]. Differences in the number of proteins and their relative concentrations between different coronas are undoubtedly linked to the physicochemical properties of NPs and the composition of biological fluids [8]. Considering the latter, it has been reported that the human plasma concentration in the incubation step affects the corona protein compositions of NPs with different surface chemistry, including commonly used nanomaterials silica and polystyrene NPs (3–80% plasma, *v*/*v*) [9,10], cationic liposomes (2.5–80% plasma) [11], gold NPs (10–80% plasma) [12] and graphene-based systems (0.1–100% plasma) [13,14]. The discrepancy in the corona protein compositions between different studies using similar NP types and plasma concentrations, however, indicates that other factors control the corona formation.

NP stability refers to the preservation of a particular nanostructure property, including crystallinity, composition, size and shape [15]. In biological media, NP aggregation that forms large and irregularly shaped clusters is one of the most common issues related to NP stability [16]. As NP aggregates are usually irregular with an arbitrary random shape and a disordered structure, the aggregation has a negative impact on both in vitro and in vivo studies. For example, it could lead to misrepresentative and irreproducible results in the cellular uptake, and could influence the biodistribution, pharmacokinetics and toxicity of NPs in an undesirable manner [16,17,18].

Cholesterol is an important component of human plasma that plays a critical role in different biological processes [19]. It is mainly transported in the bloodstream by lipoproteins, which are a heterogeneous family of particles with sizes between 10–100 nm that includes six different subspecies categorised by the density, including the high-density lipoprotein (HDL, 8–13 nm), low-density lipoprotein (LDL, 20–25 nm) and very low-density lipoprotein (VLDL, 30–80 nm) [20]. A common structure of these particles consists of a core lipid (including cholesterol, triglyceride and phospholipids) surrounded by a protein scaffold made of apolipoprotein (Apo). ApoA1 is mostly associated with HDLs, accounting for about 70% of the protein content, while apoB100 is mainly found in LDLs, and to a lesser extent, in VLDL particles [21]. The presence of both apolipoproteins and lipids has been reported in the coronas of several NPs, including copolymer NPs of *N*-isopropylacrylamide and *N*-*t*-butylacrylamide [22], silica and polystyrene NPs [23,24,25], which indicates a diverse and complex corona composition.

Our previous studies have shown that the corona formed around silica NPs greatly varied depending on the predilution of the plasma used at the incubation step [4,26]. In particular, when the NPs were incubated in a protein-deprived solution (such as 3% plasma, *v*/*v*), the corona was enriched with fibrinogen and apolipoproteins (mostly from HDLs and LDLs). In this study, we investigated if the ratio between human plasma protein mass and the NP total surface ratio (rather than the plasma protein dilution per se) played a role in determining the protein corona composition and colloidal stability on silica and carboxylated NPs. We then evaluated the impact of the LDLs and HDLs on the stability of these NPs, as these lipoproteins were present in both coronas at low plasma protein/NP ratios (low plasma concentrations), with different levels of abundance. The association of the NP aggregation and the plasma cholesterol level was further investigated in a cohort study.

## 2. Materials and Methods

### 2.1. Materials

Silica NPs (100 nm, stock concentrations of 50 mg/mL) were purchased from Kisker Biotech GmbH (Germany). Carboxylate polystyrene NPs (200 nm, stock concentrations of 25 mg/mL) were purchased from Polysciences, Inc. (Germany). HDLs and LDLs from human plasma were purchased from MyBioSource (USA). Trizma base (Tris), Glycine, Acrylamide/bis-acrylamide 40% solution, Sodium dodecyl sulphate (SDS), Ammonium persulphate (APS), N,N,N′,N′-Tetramethylethylenediamine (TEMED), phosphate buffer saline (PBS) tablets and Eppendorf LoBind microcentrifuge tubes were purchased from Sigma Aldrich (Ireland). One PBS tablet was dissolved in 200 mL of ultrapure water to obtain 10 mM of PBS (pH 7.4 at 25 °C). Trypsin Gold was purchased from Promega (UK). A blue loading buffer pack was purchased from Cell Signaling Technology (Ireland). A bicinchoninic acid assay (BCA) kit and Imperial protein stain solution were purchased from Thermo Fisher Scientific (TFS-Ireland). A Prime-Step prestained protein ladder was purchased from BioLegend (Ireland). Cholesterol measurement kits were supplied by RANDOX (UK). Human plasma from healthy donors was provided by the Irish Blood Transfusion Service (IBTS), National Blood Centre, located in Dublin. For this study, we obtained eight units of plasma from four males and four females. On the day of the collection, all units were mixed by equal volume proportions to make a biobank of average pooled plasma. The plasma was then aliquoted in 2 mL Eppendorf tubes and stored at −80 °C until use. For the cohort study, one blood sample was collected from each of the 31 volunteers who were recruited within the community via the Hugh Sinclair Unit for Human Nutrition within the University of Reading, Department of Food and Nutritional Sciences (UK). No volunteers had major underlying conditions, for example arthritis, cancer, diabetes or multiple sclerosis. Some individuals had hypertension, however none of them had a history of stroke or heart attack. Each blood sample was collected by an intravenous blood draw using a 5 mL 3.2% sodium citrate VACUETTE tube. The sample was then centrifuged at 906 RCF at 4 °C for 15 min. The supernatant plasma was collected and immediately frozen at −20 °C. Both the pooled plasma and the cohort plasma’s total protein concentrations (mg/mL) were measured with the BCA, following the manufacturer’s instructions. The plasma’s total cholesterol and HDL cholesterol concentrations were also measured based on the manufacturer’s protocols. Briefly, the total cholesterol was determined by treating the samples with cholesterol esterase and cholesterol oxidase to release hydrogen peroxide, which formed the indicator quinoneimine with 4-aminoantipyrine and phenol. The appearance of this dye was measured at 600 nm. The HDL cholesterol was measured in the same way after eliminating the non-HDL cholesterol.

### 2.2. Plasma Corona Preparation

The biomolecular corona samples were prepared by incubating NPs (silica or polystyrene) with human plasma at specific total plasma protein/NP concentration ratios. The plasma was firstly defrosted by allowing it to reach room temperature and was diluted with PBS pH 7.4 if required. Any unused plasma was discarded at the end of each day. The NPs were then incubated with the plasma solutions at 37 °C for one hour with continuous agitation. After the incubation in the plasma, the samples were centrifuged at 18,000 RCF for 10 min at room temperature to pellet the particle–protein complexes and were separated from the supernatant plasma. The pellet was then resuspended in 500 μL of PBS and was centrifuged again to pellet the biomolecular corona (1 wash) [4].

### 2.3. HDL and LDL Corona Preparation

The HDL and LDL were diluted 10 times with PBS to form the stock solutions of 261.0 mg/dL and 305.0 mg/dL, respectively. In the first experiment, the total cholesterol concentration to form a corona sample was set to 150.0 mg/dL, which has been reported to be the optimal level in normal individuals [27]. The ratios between LDL/HDL cholesterol were then varied by changing the amounts of the stock solutions to have a similar total cholesterol level to that of a 3% plasma solution (*v*/*v*) in the final volume of 500 μL, as shown in Table 1.

In the second experiment, the HDL and LDL were spiked into the incubation solution of 100 nm silica and 200 nm PS coronas (protein/NP ratios of 1.98 and 1.32, respectively). The final volume was 500 μL, which led to an incremental change of 30.5 mg/dL for the LDL cholesterol (1 LDL) in the plasma. The HDL was spiked into the incubation solution with the amount equal to an increase of 52.2 mg/dL in the plasma cholesterol level (Table 2).

To form the biomolecular corona, the NPs were incubated with the HDL, LDL and plasma for 1 h at 37 °C, before being washed once by centrifugation.

### 2.4. Corona Characterisation

DLS measurements at θ = 173° were performed using a Zetasizer Nano ZS (Malvern). Three technical replicates were measured and for each measurement, the number of runs and the duration were automatically determined. Data analysis has been performed according to standard procedures was and interpreted through a cumulant expansion of the field autocorrelation function to the second order.

The DCS experiments were performed with a CPS Disc Centrifuge DC24000, using the sucrose gradient 8–24% for silica’s samples (Analytik Ltd., UK). PVC or polystyrene calibration standards were used for each sample measurement. The time taken for spherical particles with a homogenous density to travel from the centre of the disk to the detector can be directly related with the particle size. Meanwhile, if the objects are inhomogeneous or irregular in shape, the different arrival times still allow one to distinguish between the populations, although their sizes should only be considered as an ‘apparent’ size [28].

Gel electrophoresis SDS-PAGE was performed as follows: immediately after the last centrifugation step, the corona pellet was resuspended in a protein loading buffer following the manufacturer’s instructions. The samples were boiled for 10 min at 100 °C and an equal protein amount was loaded into a 12% polyacrylamide gel. Gel electrophoresis was performed at a constant voltage of 120 V for about 60 min each until the proteins neared the end of the gel. The gels were stained in the protein stain solution, following the manufacturer’s guide. The gels were scanned using the Amersham Imager 600 (GE Healthcare Life Sciences, Ireland). The densitometry was performed with ImageJ version 1.53c (Fiji package version 2.1.0).

### 2.5. Proteomic Mass Spectrometry

Liquid chromatography coupled with mass spectrometry (LC-MS/MS) was performed on a Dionex UltiMate3000 nanoRSLC coupled in-line with an Orbitrap Fusion Tribrid mass spectrometer (TFS). Briefly, the peptide samples were loaded onto the trapping column (PepMap100, C18, 300 μm × 5 mm, 5 μm particle size, 100 Å pore size; TFS) for 3 min at a flow rate of 25 μL/min with 2% (*v*/*v*) acetonitrile and 0.1% (*v*/*v*) trifluoroacetic. The peptides were resolved on an analytical column (Acclaim PepMap 100, 75 µm × 50 cm, 3 µm bead diameter column; TFS) using the following binary gradient; solvent A (0.1% (*v*/*v*) formic acid in LC-MS grade water) and solvent B (80% (*v*/*v*) acetonitrile, 0.08% (*v*/*v*) formic acid in LC-MS grade water) using 3–50% B for 45 min, 50–90%B for 5 min and holding at 90% B for 5 min at a flow rate of 300 nL/min before returning to 3% B. MS1 spectra were acquired over *m*/*z* 380–1500 in the Orbitrap (120 K resolution at 200 *m*/*z*), and automatic gain control (AGC) was set to accumulate 4 × 10^5^ ions with a maximum injection time of 50 ms. A data-dependent tandem MS analysis was performed using a top-speed approach (a cycle time of 3 s), with precursor ions selected in the Quadrupole with an isolation width of 1.6 Da. The intensity threshold for fragmentation was set to 5000 and included charge states 2^+^ to 7^+^. Precursor ions were fragmented in the Orbitrap (30 K resolution at 200 *m*/*z*) using Higher energy Collision Dissociation (HCD) with a normalised collision energy of 28% and the MS2 spectra were acquired with a fixed first *m*/*z* of 110 in the ion trap. A dynamic exclusion of 50 s was applied with a mass tolerance of 10 ppm. The AGC was set to 5 × 10^4^ with a maximum injection time set at 300 ms.

Protein identification and quantification were performed with Maxquant, version 1.6.17.0 [29]. Using the Andromeda search engine, the MS/MS spectra were searched against the forward and reverse human Uniprot sequence database (https://www.uniprot.org, accessed on 27th December 2021). Cysteine carbamidomethylation was set as the fixed modification while the variable modifications included N-terminal acetylation and methionine oxidation. For both the protein and peptide levels, the FDR thresholds were set to 0.01 and only peptides with an amino acid length of seven or more were considered. The search filtrations were performed using a standard target-decoy database approach. Other important search parameters included a value of 0.02 Da for MS/MS mass tolerance, a value of 10 ppm for peptide mass tolerance and tolerance for the occurrence of up to two missed cleavages. The LFQ was restricted to proteins identified with at least two unique peptides. Additionally, for a protein to be considered valid, two peptide ratios were needed.

A bioinformatic analysis was performed with Perseus software, version 1.6.5.0 [30]. For the pooled silica corona dataset, log_2_ (summed intensity) was used to rank the proteins, while the log_2_ LFQ intensity was used for the protein corona comparison. The imputation of the missing values was conducted by random selection using a normal distribution with a negative shift of 1.8 standard deviations from the mean and with a width of 0.3 standard deviations. These log_2_ LFQ intensities values for all the proteins were then used for heatmap presentations (after z-scoring) and analyses.

### 2.6. Statistics and Data Plotting

Statistical analysis and data plotting were mainly performed in SPSS (version 26) and GraphPad Prism (version 9). The normality of the data was checked with the Shapiro–Wilk test. A correlation analysis was then performed separately for the normally and non-normally distributed datasets, using Pearson and non-parametric Spearman methods, respectively. The *p*-values were considered significant if they were <0.05. To evaluate the performance of the classification schemes with two groups, a receiver operating characteristic (ROC) analysis was conducted with the SPSS software. Significance was achieved if the nonparametric asymptotic significance was <0.05, with the null hypothesis being that the true AUC = 0.5. The other data were analysed and plotted with Excel (Office 2016).

## 3. Results and Discussion

### 3.1. Protein/NP Ratio Determines the Main Protein Corona Pattern

Several studies have shown the dependence of protein corona compositions on biological fluid concentrations, which is valid for different types of NPs [9,10,11,13,14]. We hypothesised that in these cases, the protein corona composition was controlled by a more comprehensive factor, which is the protein mass/total NP surface area ratio. The ratio between the protein mass and total NP surface can be calculated using the following equation:$$\mathrm{Ratio}=\frac{\mathrm{Protein}\;\mathrm{mass}\;\left(\mathrm{mg}\right)}{\mathrm{Total}\;\mathrm{NP}\;\mathrm{surface}\;\mathrm{area}\;\left(m^{2}\right)}=\frac{\mathrm{Cp}\;\left(\frac{\mathrm{mg}}{\mathrm{mL}}\right)\;.\;r\;\left(\mathrm{nm}\right)\;.\;d\left(\frac{g}{\mathrm{mL}}\right)}{3\;.\;\mathrm{Cn}\;\left(\frac{\mathrm{mg}}{\mathrm{mL}}\right)}$$
where r is particle radius (nm); d is the particle density (g/mL); and Cp and Cn are the concentrations of the human plasma total protein and particles used in the incubation step, respectively.

As the particle’s radius and density are characteristic of the NPs in use, the concentration ratio (Cp/Cn) was used to simplify the calculation for 100 nm silica NPs. The particles had a hydrodynamic size of 111.0 ± 1.1 nm and were monodispersed with a PdI of 0.021 ± 0.019. The biomolecular soft coronas of the silica NPs were prepared by incubating the particles (1 mg/mL) with pooled human plasma from eight healthy donors (a total protein concentration of 66 mg/mL) at three different plasma/NP concentration ratios: 1.98, 26.4 and 52.8, followed with one centrifugal wash. The ratios were calculated from the total plasma protein concentrations obtained by the BCA assay and the corresponding plasma concentrations (*v*/*v*) are shown in Figure 1. All the corona samples were colloidally stable, especially at the middle ratio of 26.4 (Appendix A).

In terms of the corona protein profile, the protein compositions of the silica coronas at these three protein/NP ratios were quite distinct (Figure 1—Box A). At the low ratios, the corona was enriched with the triple bands at 48, 54 and 66 kDa, and a band at 28 kDa, which were of fibrinogen and ApoA1, respectively [26]. Increasing the ratio from 1.98 to 52.8 displaced these proteins. The triple band fibrinogen diminished considerably at 26.4 while ApoA1 was only replaced after doubling the ratio. Similar protein patterns could be obtained by decreasing the NP concentration from 13.3 to 0.5 mg/mL while keeping the plasma concentration at 40% (*v*/*v*) (Box B). The same corona protein pattern was obtained when keeping the plasma protein/NP ratio at 1.98 (Box C), which demonstrates that the silica corona composition was dependent on the protein/NP ratio.

Previous studies have only reported the corona protein changes when incubating NPs at different biological fluid concentrations and one of them exploited it for proteomic biomarker discovery, which emphasised the dilution factor as the key to obtain the corona of interest [14]. In this work, the importance of the protein/NP surface ratio in controlling the corona protein composition was demonstrated. For corona-based biomarker discovery in particular, the biomolecular coronas could be more easily integrated into a low-volume, high-throughput setup, for example a plate-based protocol, without compromising the signal. Furthermore, using the protein mass/NP surface ratio that factors in the NPs’ properties would also facilitate a comparison between different studies and interpretations of the available data, which is important for understanding the impact of the biomolecular corona on therapeutic NPs.

### 3.2. Low Plasma Protein/NP Concentration Ratio Induces Silica Corona Aggregation

The biomolecular coronas formed at low protein concentrations could potentially lead to aggregation [31,32,33], which likely links to the relative abundances between the proteins and NPs. As a result, the size distribution of the silica coronas was further investigated. While the DLS results indicate a monodispersed corona at the ratio of 1.98 (Appendix A), the DCS data show that the silica corona consisted of two populations with small different densities (sizes) (Figure 2A). Although being a simple and quick benchtop method for size measurement, the DLS is known to not distinguish NP populations with close sizes [34]. The silica particles (1 mg/mL) were then incubated with the plasma at different concentrations around the 3% plasma condition (ratio = 1.98), from 0.5 to 30% (*v*/*v*). The corresponding plasma/NP concentration ratios were between 0.33 and 19.8. It can be seen that the size distribution was dependent on the protein/NP ratios, where lowering the protein/NP ratio induced more aggregation and a wider size distribution (Figure 2B). Meanwhile, the opposite could be seen when increasing the ratio to 3.3, in which the second NP population had a smaller and narrower size distribution. An increase in the samples’ PdI when decreasing the protein/NP ratios can also be seen in Appendix A.

SDS-PAGE was used to observe the corona protein changes in this ratio range, which could probably shed light on the observed NP stability (Figure 3A). Fibrinogen was found to be enriched in all the coronas formed at a protein/NP ratio below 3.3. Interestingly, at the ratios below 1.98, high MW protein bands above 175 kDa became stronger, which indicates a greater enrichment of ApoB100. The densitometry analysis revealed a relative intensity increase from 11.2 to 17.0% for these bands when decreasing the protein/NP ratio from 1.98 to 0.33 (Figure 3B). Meanwhile, the apoA1 band at 28 kDa remained mostly unchanged between 14.7 and 15.1%.

The colloidal stability of NPs in biological fluids has been an active topic of research since the protein corona gained significant attention decades ago [35,36]. While it could be difficult to predict how protein adsorption affects the colloidal stability of NPs due to its dependence on various factors, the total protein/NP surface ratio appeared to play an important role and a low ratio could induce NPs’ aggregation even with highly complex biological fluids such as human plasma. The decrease in colloidal stability could be observed by increasing the hydrodynamic sizes and PdI in the DLS, and by the DCS. The silica corona was enriched with plasma fibrinogen and lipoproteins, mainly HDLs and LDLs. While fibrinogen has been reported to potentially induce NP aggregation [31], no studies have yet determined if these lipoproteins and plasma cholesterol are linked to corona aggregation. Therefore, the associations between the presence of HDLs and LDLs and plasma cholesterol with the silica corona stability were evaluated.

### 3.3. LDL Induces Silica Corona Aggregation

The HDLs and LDLs extracted from human plasma were used to investigate the impact of cholesterol on the colloidal stability of the silica NPs. The lipoproteins were diluted in the PBS pH 7.4 to obtain the stock solutions. These bio-vessels had the main hydrodynamic sizes of 16.0 and 47.3 nm for the HDL and LDL, respectively (Appendix A). The SDS-PAGE gel of the HDL and LDL is shown in Appendix A. The HDL had the doublet bands at about 28 and 54 kDa, while the major protein bands of the LDL stayed in the high MW range above 130 kDa. The lipoproteins were first incubated with the NPs at different LDL/HDL cholesterol ratios while keeping the total cholesterol constant at 150 mg/dL, which has been reported to be the normal cholesterol level [27]. From the DLS analysis, it can be seen that the stability of the silica NPs decreased with increasing LDL/HDL ratios, and both the size and PdI reached the plateau at the ratio of two (Figure 4A). Beyond this point, the particle highly aggregated with the PdI above 0.4.

The lipoproteins were then spiked into the pooled plasma incubation solution (protein/NP ratio = 1.98) to mimic the increase in the plasma HDL and LDL cholesterol levels. The spiked LDL plasma coronas had increased sizes and wider DCS size distributions, which was similar to the size distribution of the plasma corona formed at a lower protein/NP ratio of 0.66 (Figure 4B). Figure 4C demonstrates that introducing the LDL caused silica corona aggregation, with increases in both the hydrodynamic sizes and PdI values, while the presence of the HDL, to some extent, reduced the degree of aggregation. The DCS analysis did not reveal any other major peaks from the corona sample, which could be originated from the free HDL and LDL pelleted along with the coronas by centrifugation. Overall, the results indicate that the LDL could decrease the colloidal stability of the silica coronas, while the presence of the HDL slightly alleviated that effect.

### 3.4. The Silica Corona Stability Is Associated with Plasma Cholesterol Level

As HDLs and LDLs are the main lipid carriers in the bloodstream, particularly for cholesterol and triglyceride, we investigated the association between the silica corona stability and the plasma cholesterol levels of 31 volunteers (18 males and 13 females) with a median age of 62.0 (54.0–65.5) whose cholesterol concentrations are shown in Table 3. The study was designed with the purpose of establishing correlations between the corona sizes and PdI obtained with the DLS and the measured cholesterol concentrations, particularly the cholesterol ratio taking into account both the plasma HDL and LDL cholesterol levels.

Both the total cholesterol and especially the cholesterol ratio were positively correlated with the increases in size and the PdI, which indicates a detrimental effect of the non-HDL cholesterol to the stability of the silica NPs (Figure 5A,B). Meanwhile, the downward trend was observed with the HDL cholesterol level, which rendered the silica corona more stable (Figure 5C). Overall, the size and PdI had a stronger correlation with the cholesterol ratio than the HDL or LDL cholesterol alone, and the strongest correlation was observed between this ratio and the PdI, with a coefficient of determination of R^2^ = 0.587 and a correlation coefficient of r = 0.77 (Table 4).

The plasma samples were then split into two groups based on their coronas’ PdI values: below 0.25 which was considered to be ‘stable’ and above 0.25 (unstable). The ROC analysis shows that this PdI-based stability of the silica coronas could be distinguished by both the total plasma cholesterol and plasma HDL cholesterol concentrations, with an AUC of 0.724 and 0.647, respectively. However, combining them together in the form of a cholesterol ratio provides good prediction power with an AUC of 0.814 (Figure 5D). Overall, this analysis confirms the strong relationship between the cholesterol ratio and the degree of NP aggregation, which is likely dependent on the relative abundances between the HDL and LDL cholesterol. Furthermore, it also demonstrates that the biomolecular corona characteristics could be correlated with human plasma’s physiological properties, which can be exploited for biomarker discovery. As plasma cholesterol level could vary depending on lifestyle and medication use [37], biomolecular corona-based biomarker discovery studies should pay attention to any influence of this factor on the corona stability, which could affect the reliability of the identified biomarkers.

To further establish the link between the plasma cholesterol, the lipoprotein abundance in the silica corona and the colloidal stability, proteomic MS was used to analyse three coronas obtained from the plasma samples with low, medium and high cholesterol ratios (1.89, 2.72 and 5.87, respectively). The label-free quantification (LFQ) shows noticeable differences between the low and high ratio coronas (Figure 6A). Compared to the high ratio corona, the low ratio one was enriched with apoA4, apoF and especially apoA1 present in the HDL, while it had a lower level of apoA2 and the LDL-related apoB100. With increasing plasma cholesterol ratios, more apoB100 was present in the silica corona while the presence of apoA1 decreased progressively (Figure 6B). As plasma apoA1 had a strong affinity to the silica surface in the low protein/NP conditions, it was of little surprise that the level of this protein in the corona could reflect that in the plasma incubation solution. Interestingly, the level of apoA2 increased along with apoB100 in the high cholesterol ratio corona (Figure 6A), which could imply the presence of chylomicrons or VLDL. The DCS analysis shows that the silica NPs formed larger aggregates, particularly in the second peak, with an increasing cholesterol ratio, and a wider size distribution (Figure 6C). Overall, the plasma cholesterol likely affected the stability of the silica NP via inducing the variation of lipoprotein levels in the corona.

To verify if the abundance of lipoproteins in the biomolecular corona could affect this association, a similar study was performed on 200 nm carboxylated polystyrene NPs. These particles had a more ‘hydrophobic’ surface and, more importantly, were capable of enriching both the HDL and LDL less than the silica NPs at the plasma/NP ratio of 1.32 (Appendix A). 100 nm polystyrene NPs were much less stable than 200 nm polystyrene NPs at similar plasma/NP concentration ratios (data not shown); hence, they were not used for this purpose. The results demonstrate that the stability of the polystyrene corona also decreased when increasing the LDL/HDL ratio (Appendix A) and in the cohort plasma samples with high cholesterol ratios (Appendix A). However, the correlation between the corona stability and plasma cholesterol level was weaker than that observed with the silica NP. When being incubated with the plasma HDL and LDL, the polystyrene NPs formed aggregates at a slower rate, with the corona PdI reaching a plateau earlier and staying below 0.4, and the hydrodynamic sizes went up gradually (Appendix A). Moreover, the cohort study shows much a lower correlation coefficient (and *p*-value) between cholesterol ratio and this NP stability (Appendix A). As a result, the polystyrene system appeared to be less susceptible to the presence of LDLs, which is likely due to the lower enrichment of this lipoprotein in the corona. Interestingly, the polystyrene NPs (200 nm, density = 1.06 g/cm^3^) at the protein/NP concentration ratio of 1.32 were in a more ‘protein-deprived’ condition than the silica NPs at the ratio of 1.98 (100 nm, density = 2 g/cm^3^), as were the highly enriched plasma fibrinogen (Appendix A), and both factors have reportedly caused NP aggregation [32,38]. Hence, the results indicate that the LDL cholesterol was another factor controlling the observed aggregation. In fact, the difference in stability between the two types of NPs could be well explained by the difference in their corona LDL abundances.

The association between the plasma cholesterol level and aggregation of both the silica and polystyrene NPs could be confounded by the high enrichment of fibrinogen at a low plasma concentration. Therefore, a preliminary experiment was conducted with silica coronas formed at a protein/NP ratio of 52.8 (80% plasma, *v*/*v*), which carried much less of this protein (as well as both HDLs and LDLs) [26]. Appendix A shows a weaker correlation between the cholesterol ratio and this corona’s PdI, in comparison with the silica corona formed at the ratio of 1.98, which is likely due to the lower presence of both the LDLs and HDLs. Some plasma coronas with high cholesterol ratios were also prepared with polystyrene NPs at high ratios of 35.2 (80% plasma, *v*/*v*) and all of them were very stable, which is in line with the observed low influence of the cholesterol level on the stability of this particle (Appendix A).

While centrifugation is an established method for preparing the biomolecular corona, it could introduce artefacts, including the co-isolation of bio-NPs such as lipoproteins [21]. However, the low sample background contributed to the clear observation of the apolipoprotein abundance differences between the silica and polystyrene coronas prepared with the same method, as well as the low correlation observed with the silica coronas formed at the 80% plasma concentration (*v*/*v*). Plasma fibrinogen is an important protein in the coagulation process [39] and has been reported in various biomolecular coronas [40]. The impact of this protein concentration on the stability of NPs will be investigated in the future.

## 4. Conclusions

In this study, the impact of the relative abundances between plasma protein and silica NPs on the biomolecular corona formation was demonstrated. The association between the plasma cholesterol and corona colloidal stability was established in a cohort analysis, particularly between the plasma cholesterol ratio and the PdI obtained with DLS. The level of the plasma cholesterol was reflected in the silica corona enrichment of apoA1 and apoB100, which, in turn, induced the particle’s aggregation, depending on the balance between the HDL and LDL. While the HDL appears to stabilise the coronas, its effect was not as strong as the destabilisation induced by the LDL. In comparison to the silica and polystyrene NPs, while fibrinogen was enriched heavily by both of them, the difference in the cholesterol level, particularly associated with the HDL and LDL, could explain the stronger tendency of the silica NP to aggregate. This work highlights that biomolecular corona studies should characterize the NP colloidal stability thoroughly, particularly in conditions where the NPs tend to form the corona with the LDL.

## Figures and Tables

**Figure 1 nanomaterials-12-02661-f001:**
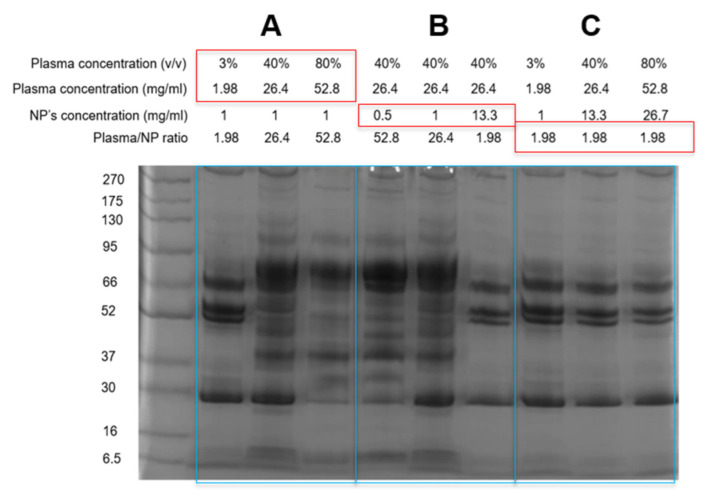
SDS-PAGE gel showing the protein patterns of silica coronas. The concentrations of the particles and plasma proteins were varied around three protein/NP concentration ratios: 1.98, 26.40 and 52.80. (**A**,**B**) The protein composition of the silica corona strongly depended on the protein/NP ratio. (**C**) Diluting the incubation solution, for example from 80% to 3% while keeping the same plasma/NP ratio of 1.98, did not affect the corona protein composition.

**Figure 2 nanomaterials-12-02661-f002:**
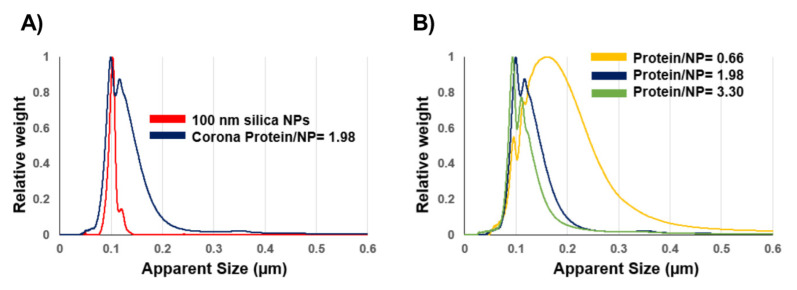
Decreasing the plasma protein/NP concentration ratios induces aggregation of 100 nm silica NPs. (**A**) DCS size distribution of 100 nm silica and its corona at the protein/NP ratio of 1.98. (**B**) The sample colloidal stability was improved (narrower size distribution) when increasing the plasma/NP ratio.

**Figure 3 nanomaterials-12-02661-f003:**
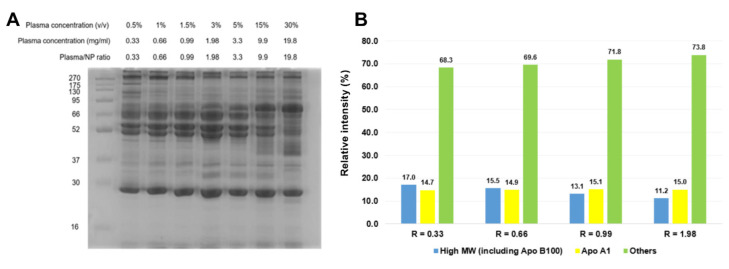
(**A**) Protein profile of silica coronas at different plasma protein/NP ratios. The NP concentration was set at 1 mg/mL while the plasma concentration varied from 0.33 to 19.8 mg/mL. (**B**) When increasing the plasma protein/NP ratio (R) from 0.33 to 1.98, the abundance of high MW proteins (>175 kDa) decreased.

**Figure 4 nanomaterials-12-02661-f004:**
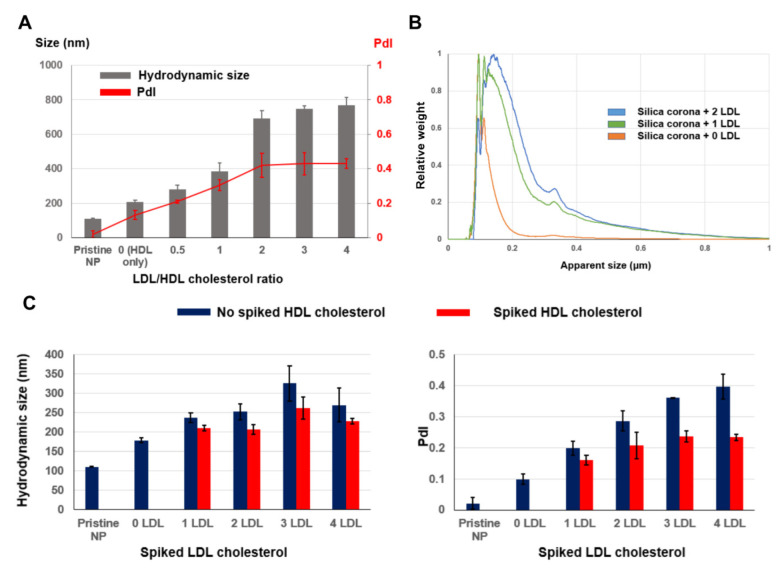
DLS and DCS sizes of silica corona formed with human plasma HDL and LDL. (**A**) Both hydrodynamic size and PdI of lipoprotein silica coronas increased with the increasing LDL/HDL ratios. The total amount of HDL and LDL cholesterol were calculated to be equal to 150 mg/dL while the ratios between the LDL and HDL cholesterol levels were varied. (**B**) The spiked LDL plasma coronas (protein/NP ratio = 1.98) had wider size distributions. (**C**) Both corona hydrodynamic size and PdI increased with increasing amounts of added LDL, while spiked HDL reduced the degree of aggregation. One LDL equalled an incremental change of 30.5 mg/dL in total plasma cholesterol, while HDL was spiked into the incubation solution with the amount equal to an increase of 52.2 mg/dL in cholesterol level. Error bar: SD of the mean (*n* = 3).

**Figure 5 nanomaterials-12-02661-f005:**
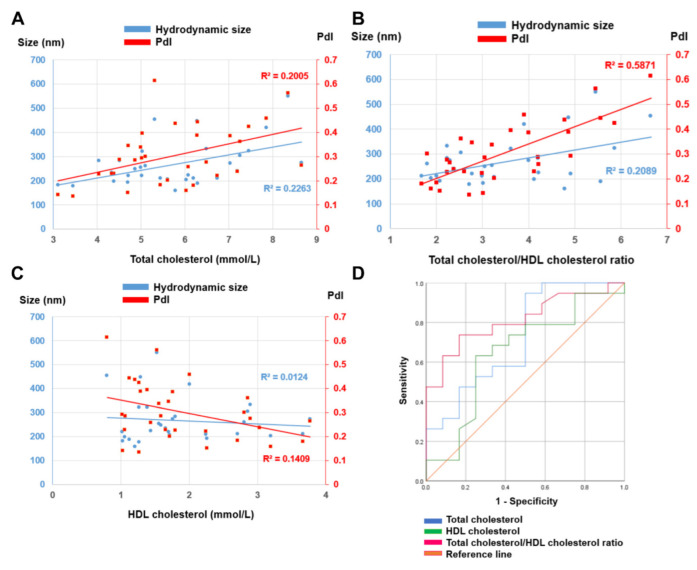
Association between the plasma cholesterol level and the stability of silica coronas. (**A**) Positive correlations between total cholesterol concentration and both hydrodynamic size and PdI. (**B**) Positive correlations between cholesterol ratio and both hydrodynamic size and PdI. The highest R^2^ was observed between the cholesterol ratio and PdI. (**C**) Negative correlations between HDL cholesterol and both hydrodynamic size and PdI. (**D**) ROC curves of total cholesterol, HDL cholesterol and cholesterol ratio for classifying the silica corona samples based on their PdI (below or above 0.25). The highest AUC was achieved with the cholesterol ratio (0.814).

**Figure 6 nanomaterials-12-02661-f006:**
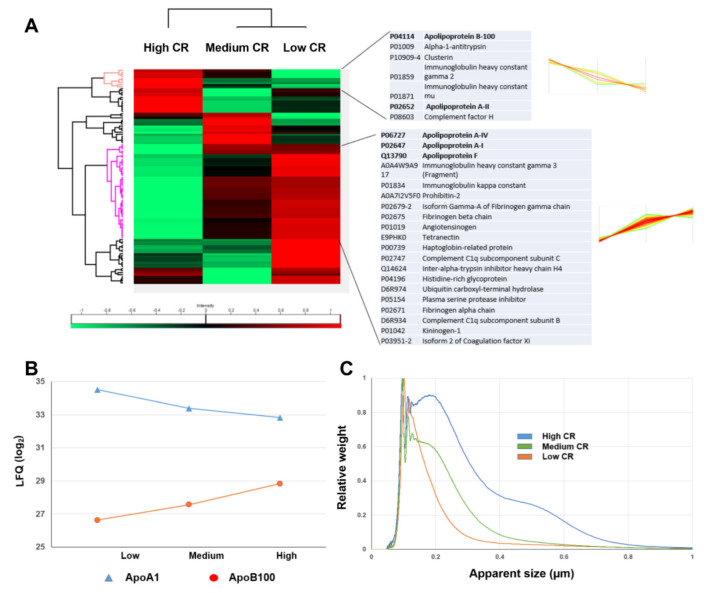
Protein corona composition changed according to the plasma cholesterol levels. (**A**) Heat map shows the LFQ intensities (low: green, high: red) of silica coronas formed in plasma samples with three different cholesterol ratios: low CR (1.89), medium CR (2.72) and high CR (5.87). Protein clusters with consistent increases or decreases from low to high cholesterol ratios are highlighted. The intensity profiles of some protein members are shown. (**B**) ApoA1 levels in the coronas decreased with increasing cholesterol ratio while the opposite was observed with ApoB100. (**C**) DCS size distributions of these three corona samples show the most aggregation in the high cholesterol ratio sample.

**Table 1 nanomaterials-12-02661-t001:** The amount of HDL and LDL stock solution used in the incubation step to form lipoprotein silica corona.

LDL/HDL Ratio	Calculated HDL Cholesterol (mg/dL)	HDL Stock Volume (μL)	Calculated LDL Cholesterol (mg/dL)	LDL Stock Volume (μL)
0	136.4	7.8	0	0
0.5	81.5	4.2	40.5	2
1	58	3.3	58	2.8
2	37	2.1	74	3.6
3	27	1.6	81	4
4	21	1.2	84	4.2

**Table 2 nanomaterials-12-02661-t002:** The amount of HDL and LDL stock solution spiked into the plasma incubation solution. 1 LDL equalled to a change of 30.5 mg/dL LDL cholesterol; 1 HDL equalled to a change of 52.2 mg/dL HDL cholesterol.

Sample	Plasma (μL)	HDL Stock Volume (μL)	LDL Stock Volume (μL)
1 LDL	15	0	1.5
2 LDL	15	0	3
3 LDL	15	0	4.5
4 LDL	15	0	6
1 HDL + 1 LDL	15	3	1.5
1 HDL + 2 LDL	15	3	3
1 HDL + 3 LDL	15	3	4.5
1 HDL + 4 LDL	15	3	6

**Table 3 nanomaterials-12-02661-t003:** Descriptive information for 31 volunteers. The continuous variables total plasma cholesterol, HDL cholesterol and cholesterol ratio are shown as medians and interquartile ranges.

Cholesterol Level	Plasma Sample (*n* = 31)
Total plasma cholesterol in mg/dL (median [IQR])	216.9 (185.6–256.0)
HDL cholesterol in mg/dL (median [IQR])	64.2 (49.1–95.9)
Cholesterol ratio (median [IQR])	3.1 (2.3–4.2)

**Table 4 nanomaterials-12-02661-t004:** Correlation coefficients (r) of the plasma cholesterol level (total cholesterol, HDL cholesterol and cholesterol ratio) and the silica corona’s colloidal stability (DLS size and PdI). Pearson correlation and Spearman correlation coefficients were calculated for normally distributed and non-normally distributed data, respectively. *: Pearson correlation coefficients. *p* < 0.05 are bolded.

	Total Cholesterol		HDL Cholesterol		Cholesterol Ratio	
	r	*p*	r	*p*	r	*p*
**Hydrodynamic size**	0.43	**0.016**	0.08	0.66	0.23	0.214
**PdI**	0.45 (*)	**0.012**	−0.33	0.071	0.77 (*)	**<0.0001**

## Data Availability

The data presented in this study are available on request from the corresponding author.

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
