# Peer review of "Biomolecular Corona Stability in Association with Plasma Cholesterol Level"

_nanomaterials, 2022, doi:10.3390/nano12152661_

Round 1

Reviewer 1 Report

The authors characterized biomolecular corona formed on the silica NP (nanoparticle) surface after contact with biological fluids. Their study shows the crucial effect of the ratio between plasma protein mass and NP total surface on the stability of the nanoparticle and, therefore, on their size distribution. 

Furthermore, the effect of the cholesterol concentration on the corona stability was investigated, showing that LDL cholesterol has a detrimental impact on the stability of the NPs, and an increase in HDL cholesterol concentration slightly mitigates such an effectThe cause has been identified as the modification of the corona composition after the change in cholesterol concentration. The authors proved their thesis by also investigating the biomolecular corona on carboxylated polystyrene NPs, which show an attenuated effect of the phenomena. 

The study is well-conceived and executed. It shows experimental evidence of the conclusions reached.

I suggest the authors complement the work with additional measurements to obtain a “calibration curve” of the protein mass and NP total surface ratio vs particle stability. Is there an optimal ratio value at which the NP surface is fully saturated, and the NPs do not tend to aggregate? Can you use thermodynamic arguments to explain it?

Reviewer 2 Report

The paper “Biomolecular corona stability in association with plasma cholesterol level” by Duong N. Trinh et.al is dedicated to a cohort study on plasma samples from individuals with known cholesterol levels, to highlight the association between the plasma cholesterol and corona colloidal stability. Moreover, the authors demonstrated the impact of the relative abundances between plasma protein and silica NPs on the biomolecular corona formation.

The paper is well written and it shows that the authors did a lot of work. However, I have some mild comments and suggestions:

1. In the Introduction section, at page 2, row 50 to 59, the authors talked about different NPs size measurement techniques. I suggest that the authors should remove this part, as these techniques are well known.

2. In the Section 2. Materials and methods- 2.2. Plasma corona preparation, the authors mentioned that “NPs were allowed to incubate with the plasma solutions at 37°C for one hour with continuous agitation”. Is there a specific protocol for the incubation period?

3. As referred to the incubation period of NPs+plasma solutions for 1 hour and to the 3. Results and discussion - 3.1. Protein/NP ratio determines the main protein corona pattern section, I will like to know if there exist some effects on the incubation period on the formation of the corona?

4. In the 3. Results and discussion- 3.4. The silica corona stability is associated with plasma cholesterol level section, I see that the median age of the volunteers is 62. To my knowledge, cholesterol levels depend on the health, age, etc of an individual. How was the group selected?  

5. Figure 4, B), the “name” of the Y axis is missing.

Author Response

Please find file attached 
